# Divide and Conquer: Reliable Multi-View Evidential Learning for Deepfake Detection

Xiaolu Kang [1]  Zhongyuan Wang [1]  Jikang Cheng [2]  Baojin Huang [3]  Zhanhe Lei [1]  Gang Wu [4]  Qin Zou [1]
Qian Wang [5]

## Abstract

With the evolution of generative models, deepfakes have achieved near-perfect semantic realism, leaving forensic traces only in subtle structural anomalies. However, existing single-view paradigms often fail to generalize, as dominant semantic features overwhelm subtle artifact cues within entangled representations. This imbalance leads to overconfident yet brittle predictions—a phenomenon we term the *Semantic Masking Effect*. To address this challenge, we propose a reliable framework called *Divide-and-Conquer Multi-View Evidential Learning* (**DiCoME**) for Deepfake Detection. In the "Divide" phase, we employ Geometric View Purification to decompose the entangled representation space through principled geometric projection. This process suppresses semantic interference within artifact-sensitive representations, forming the foundation for decorrelated yet complementary semantic and artifact views. In the "Conquer" phase, we leverage Uncertainty-Aware Evidential Learning to synthesize these distinct views. By explicitly modeling the "epistemic conflict" between semantic and artifact cues, this mechanism provides calibrated uncertainty estimates instead of forcing rigid deterministic decisions. Extensive experiments across multiple benchmarks demonstrate that our method consistently outperforms existing approaches in generalization performance, while providing reliable uncertainty estimation for trustworthy deepfake detection. Code is available at https://github.com/kxl0825/DiCoME.git.

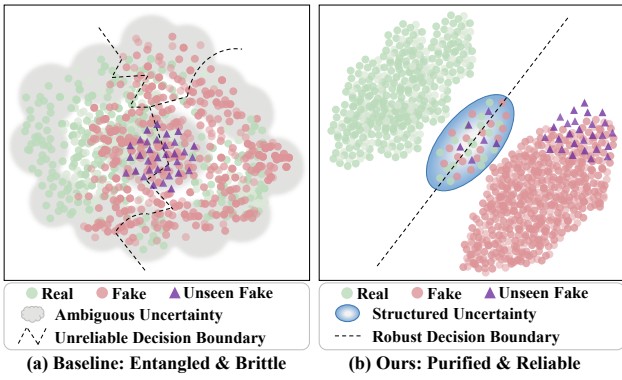

Real  Fake  ▲ Unseen Fake
Ambiguous Uncertainty
⌄⌄ Unreliable Decision Boundary

Real  Fake  ▲ Unseen Fake
Structured Uncertainty
---- Robust Decision Boundary

**(a) Baseline: Entangled & Brittle**  **(b) Ours: Purified & Reliable**

*Figure 1.* Feature entanglement vs. Our solution. (a) Dominant semantic features overshadow subtle artifact cues, causing unseen fake samples to entangle with real data. This leads to ambiguous uncertainty and unreliable decision boundaries that fail on new attacks. (b) Artifact features are isolated from semantic content. Hard unseen samples are captured in a structured uncertainty region, enabling robust boundaries and superior generalization.

## 1. Introduction

The democratization of high-fidelity synthesis via GANs (Goodfellow et al., 2014) and Diffusion Models (Ho et al., 2020) has elevated deepfakes from a technological curiosity to a serious security threat, enabling large-scale misinformation and financial fraud (Sunil et al., 2025). While generative models continue to advance at a remarkable pace, deepfake detection methods still suffer from a fundamental generalization bottleneck, particularly when facing unseen manipulation techniques and cross-domain scenarios (Lin et al., 2024; Zhang et al., 2025a). This gap highlights the urgent need for detection methods that capture universal and transferable forensic cues, rather than overfitting dataset-specific patterns (Chandra et al., 2025).

A fundamental challenge to robust generalization arises from the intrinsic conflict between high-level semantic realism and low-level structural anomalies. Most existing models adopt a single-view holistic processing paradigm (Rossler et al., 2019), employing monolithic backbones to indiscriminately encode both semantic content and forgery traces into an entangled representation. However, visual backbones are typically optimized for semantic invariance, causing high-magnitude identity features to inherently dom-

[1]School of Computer Science, Wuhan University, Wuhan, China [2]School of Integrated Circuits, Peking University, Beijing, China [3]School of Information, Huazhong Agricultural University, Wuhan, China [4]College of Cyber Security, Tarim University, Alaer, China [5]School of Cyber Science and Engineering, Wuhan University, Wuhan, China. Correspondence to: Zhongyuan Wang <wzy_hope@163.com >.

*Proceedings of the 43$^{rd}$ International Conference on Machine Learning*, Seoul, South Korea. PMLR 306, 2026. Copyright 2026 by the author(s).

inate the representation. Consequently, dominant semantic signals tend to overwhelm weak artifact cues within this entangled feature space, a phenomenon we term the *Semantic Masking Effect*. As illustrated in Fig. 1(a), such feature entanglement forces models to learn unreliable decision boundaries that lead to baseless overconfidence on realistic fakes. In contrast, an ideal detector should explicitly suppress semantic interference to isolate artifact-sensitive evidence. As shown in Fig. 1(b), our solution yields a robust decision boundary and establishes a structured uncertainty zone, enabling reliable decision-making even when the model faces unknown manipulation techniques.

To address these challenges, we propose *Reliable Multi-View Evidential Learning for Deepfake Detection*, governed by a "Divide-and-Conquer" strategy. Departing from monolithic single-source reliance, we establish two complementary expert perspectives collaborating within a trustworthy method: **View I: The Semantic View (Content Expert).** Leveraging the universal priors of Vision Foundation Models (e.g., CLIP (Radford et al., 2021)), this view evaluates high-level plausibility based on semantic consistency, answering "What you see". **View II: The Artifact View (Structure Expert).** Grounded in intrinsic image formation laws, this view scrutinizes the low-level signal domain to expose manifold irregularities, addressing "How it is generated". However, naive fusion strategies of these two views overlook feature entanglement, rendering the two experts highly correlated. Consequently, the "Structure Expert" merely echoes the "Content Expert" rather than offering distinct corroboration. Such coupling not only undermines genuine multi-view complementarity but also breeds overconfidence in erroneous predictions due to the redundant accumulation of information.

Our core insight establishes geometric orthogonality as an inductive bias encouraging view-level decorrelation and complementarity. We rigorously define artifacts not as noise, but as the orthogonal complement of the semantic space. By enforcing orthogonal projection as a geometric hard constraint, we empower the "Structure Expert" to mine manifold anomalies immune to semantic interference. We further introduce Multi-view Evidential Learning for decision-level fusion. Treating views as decorrelated yet complementary sources, this paradigm ensures that even faint orthogonal anomalies generate distinct evidence of fake. This mechanism effectively breaks the gradient monopoly of strong semantics, granting subtle artifact signals equal voting rights in the final decision. Consequently, the final decision transcends simple probability averaging by synthesizing uncertainty-aware evidence reports. Crucially, inconsistencies between views allows the model to capture "epistemic conflict" (Kendall & Gal, 2017): discrepancies between experts (e.g., Semantic-"Real" vs. Artifact-"Fake") trigger high-uncertainty warnings rather than blind

misclassifications, ensuring trustworthy generalization towards unseen attacks. In summary, our contributions can be summarized as:

- **Geometric View Purification for Deepfake Detection:** We introduce a geometric projection mechanism that constructs artifact representations within the learned orthogonal complement of the semantic space, effectively mitigating the *Semantic Masking Effect* and enabling decorrelated yet complementary semantic and artifact views.
- **Uncertainty-Aware Multi-view Evidential Fusion:** We employ Dempster-Shafer Theory for decision-level fusion, leveraging the decorrelation and complementarity of the two views. This mechanism effectively captures the "epistemic conflict" between experts, mitigating the blind overconfidence of traditional fusion methods and enabling precise trustworthiness quantification.
- **Superior Generalization and Trustworthiness:** Extensive experiments confirm that our method achieves state-of-the-art performance on cross-domain benchmarks, ensuring both robust generalization and superior reliability.

**Conflict of Interest Disclosure.** The authors declare that they have no relevant interests to disclose.

## 2. Related Work

**Deepfake Detection.** Deepfake detection has evolved from mining low-level artifacts (Tan et al., 2024; Qian et al., 2020; Ciftci et al., 2020) and modeling face manifolds (Cao et al., 2022) to improving generalization under diverse and evolving forgery distributions through multi-domain learning, explicit debiasing, progressive forgery modeling, diffusion-guided representation learning, and continual adaptation (Cheng et al., 2025a;b; 2024; Sun et al., 2024; Zhang et al., 2025b). Another line of research leverages Vision Foundation Models (VFMs) for deepfake detection. While VFM-based methods (Xu et al., 2023; Yermakov et al., 2026; Cui et al., 2025) utilize CLIP (Radford et al., 2021) for semantic consistency, they often suffer from feature entanglement, where subtle artifacts are overshadowed by semantic redundancy. Although recent works like FDML (Yu et al., 2024) attempt to distill artifacts via attention-guided fusion and data augmentation, relies on implicit soft constraints lack the geometric rigor to prevent semantic leakage. More recently, Effort (Yan et al., 2025) utilizes SVD-based orthogonal projection but suffers from limitations due to its reliance on a global rank reduction strategy. First, this enforces a uniform low-rank assumption that ignores instance-specific nuances, inevitably discarding subtle forgery traces unique to individual samples. Furthermore, the rigidity of these fixed rank constraints restricts the model's flexibility, hampering its generalization across varying data distributions. Finally, these methods mentioned above rely on deterministic inference that overlooks "epistemic conflict" (Kendall & Gal, 2017), failing to quantify predictive uncertainty.

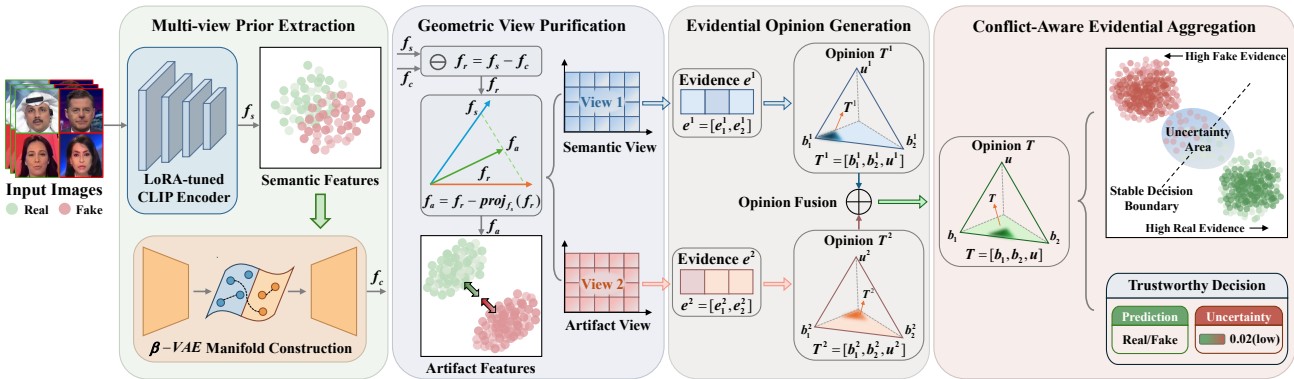

*Figure 2.* Overview of the proposed method. (1) Multi-view Prior Extraction: Semantic feature $f_s$ is established via a LoRA-tuned CLIP encoder to form the Semantic View, while a manifold-consistent feature ($f_c$) is reconstructed via a $\beta$-VAE. (2) Geometric View Purification: To generate the Artifact View constituted by $f_a$, raw residuals ($f_r = f_s - f_c$) are projected onto the semantic orthogonal complement, effectively decoupling artifacts from semantic interference. (3) Evidential Opinion Generation: Disentangled views are mapped to Dirichlet distributions to yield Subjective Opinions ($T^1, T^2$) parameterized by Belief ($b$) and Uncertainty ($u$). (4) Conflict-Aware Aggregation: Dempster-Shafer fusion integrates opinions to produce trustworthy predictions, explicitly leveraging "epistemic conflict" for uncertainty quantification.

**Multi-view Learning.** Multi-view Learning enhances robustness by integrating diverse perspectives, such as "RGB + Frequency" (Tan et al., 2024; Qian et al., 2020) or "Global + Local" (Zhao et al., 2021). However, existing approaches typically presume implicit complementarity, often overlooking critical information redundancy. For instance, semantic features inevitably encode texture cues, hindering the learning of truly disentangled representations. Furthermore, standard deterministic fusion (e.g., Softmax) fails to resolve "epistemic conflict" (Kendall & Gal, 2017): when views yield contradictory predictions, naive aggregation masks the underlying uncertainty, causing overconfidence on hard samples. In contrast, we employ Geometric View Purification to explicitly decouple artifacts via orthogonal projection and leverage Evidential Deep Learning (Sensoy et al., 2018) to quantify uncertainty, ensuring superior trustworthiness at both feature and decision levels.

## 3. Methodology

### 3.1. Motivation

Deepfake detection fundamentally entails identifying faint anomalies embedded within a strong semantic background. We formulate the feature representation of deepfake images as a superposition of a dominant semantic component $f_s$ and a subtle artifact component $f_a$. Generative models prioritize optimizing $f_s$ to achieve high realism, creating a significant magnitude disparity ($\|f_s\| \gg \|f_a\|$) in the feature space. In traditional monolithic single-view paradigms, the optimizer is biased towards capturing these dominant semantic features as a shortcut for rapid loss minimization. This triggers a *Semantic Masking Effect*, where faint artifacts are overshadowed by powerful semantic representations. Moreover, traditional feature extraction entangles semantic and artifact components, making it difficult to isolate manipulation

traces without semantic contamination. As a result, models tend to overfit semantic regularities and fail to generalize to semantically realistic yet artifactually novel attacks.

Although recent methods attempt to incorporate both semantic and artifact cues, they predominantly rely on feature-level fusion, which remains vulnerable to semantic leakage: artifact representations are still corrupted by residual semantic information. Consequently, fusion devolves into a superposition of entangled information dominated by strong semantics. More critically, the widely adopted Softmax classifier enforces a deterministic decision under the assumption of mutual exclusivity, preventing the model from expressing epistemic uncertainty when conflicting evidence arises across views, and leading to overconfident predictions on unseen attacks.

To overcome these limitations, we propose a *Reliable Multi-View Evidential Learning* framework under a "Divide-and-Conquer" strategy. Specifically, we treat artifact representations as the orthogonal complement of the semantic space and achieve inter-view decorrelation and complementarity via explicit geometric projection, thereby suppressing semantic interference by construction rather than through soft regularization. Building upon this disentanglement, we adopt Dempster–Shafer Theory to explicitly model "epistemic conflict" between views. Unlike black-box fusion, our framework detects contradictory evidence (e.g., Semantic-"Real" vs. Artifact-"Fake") and responds with elevated uncertainty, enabling robust rejection of unknown attacks in open-world deepfake detection scenarios.

### 3.2. Overview

As illustrated in Figure 2, the proposed framework follows a "Divide-and-Conquer" strategy to achieve reliable deepfake detection. In the "Divide" phase, we construct two

decorrelated yet complementary views: a *Semantic View* anchored by CLIP and an *Artifact View* obtained through our Geometric View Purification module, which enforces principled geometric decoupling to suppress semantic interference in artifact-sensitive representations. In the "Conquer" phase, we employ Uncertainty-Aware Evidential Learning to transform these decoupled features into subjective opinions (Content Expert vs. Structure Expert). Finally, we utilize Dempster-Shafer theory to synthesize these opinions, explicitly capturing "epistemic conflict" to enable robust cross-domain generalization and reliable rejection of unknown attacks.

### 3.3. Divide: Multi-View Construction

This phase constructs two geometrically decorrelated yet complementary evidence sources: a **Semantic View** anchored by high-level priors, and an **Artifact View** purified via geometric projection. By isolating artifact residuals in the orthogonal complement of the semantic subspace, we effectively decouple artifacts from semantics to provide the decorrelated and complementary evidence streams required for multi-view reasoning.

**Multi-View Prior Extraction.** To capture the intrinsic opposition between high-level semantic consistency and low-level manifold anomalies at the feature level, we construct a dual-branch feature extraction mechanism. First, visual foundation models such as CLIP encode robust and transferable semantic priors, but suffer from domain shift when directly applied to specialized forensic tasks. Full fine-tuning risks catastrophic forgetting of these general priors. To address this, we employ Low-Rank Adaptation (LoRA) (Hu et al., 2022) to efficiently adapt the CLIP image encoder to the deepfake domain. Formally, given an input image $x$, the adapted encoder extracts a semantic feature representation $f_s \in \mathbb{R}^d$:

$$f_s = \mathcal{E}_{\text{clip}}(x; \theta_{frozen}, \theta_{lora}), \tag{1}$$

where $f_s$ serves as the anchor of "semantic view", capturing identity-related and high-level semantic consistency while maintaining strong generalization capability of the pre-trained backbone.

Simultaneously, to capture non-semantic anomalies introduced during generation, we adopt a manifold-based assumption: real faces approximately reside on a compact manifold in the semantic feature space, while deepfakes deviate from it. Instead of pixel-level reconstruction, we employ a lightweight $\beta$-VAE (Higgins et al., 2017) as a manifold projector directly operating on semantic features $f_s$. By reconstructing $f_s$ into a manifold-consistent feature $f_c$, the projector captures the dominant semantic structure of CLIP representations, forming a semantic-dominant manifold that preserves high-level identity information. As semantic components correspond to the principal directions

of $f_s$, manipulation-induced non-semantic perturbations are not faithfully reconstructed and instead emerge as feature-level deviations, which are subsequently isolated via explicit orthogonal projection.

Finally, we compute the residual representation to capture these deviations:

$$f_r = f_s - f_c, \tag{2}$$

This residual encapsulates preliminary anomalies unexplained by the manifold projector. However, since semantic information may still leak into $f_r$, it is treated as a raw residual and further refined through the proposed Geometric View Purification to obtain purified artifact representations.

**Geometric View Purification.** Although the residual $f_r$ captures preliminary anomalies, it remains inherently contaminated by semantic leakage due to the limited reconstruction capacity of the manifold projector. As a result, $f_r$ exhibits non-zero correlation with the semantic feature $f_s$, violating the requirement for decorrelated and complementary evidence streams underlying multi-view evidential learning. To address this, we impose an explicit geometric orthogonality constraint to purify the artifact signal. Treating $f_s$ as the semantic direction in the feature space, we decompose $f_r$ into two components: a parallel component ($f_r^{\parallel}$), which captures collinear semantic leakage, and an orthogonal component ($f_a$), which represents purified artifact signal. According to the vector projection theorem, the semantic leakage component is quantified as:

$$f_r^{\parallel} = \text{proj}_{f_s}(f_r) = \frac{\langle f_r, f_s \rangle}{\|f_s\|_2^2} f_s, \tag{3}$$

Consequently, the purified artifact feature is isolated by subtracting this component:

$$f_a = f_r - f_r^{\parallel} = f_r - \text{proj}_{f_s}(f_r), \tag{4}$$

Geometrically, this operation projects the residual representation onto the orthogonal complement space $S^{\perp}$) of the semantic direction, ensuring $\langle f_a, f_s \rangle = 0$. By enforcing geometric decorrelation, the proposed projection suppresses the *Semantic Masking Effect* and enables $f_a$ and $f_s$ function as decorrelated evidence sources. This provides the necessary theoretical justification for subsequent Dempster–Shafer fusion to reliably distinguish between consistent and conflicting evidence across views.

### 3.4. Conquer: Uncertainty-Aware Evidential Learning

This phase synthesizes the decoupled views into a trustworthy global decision. We leverage Evidential Deep Learning to transform features into subjective opinions and fuse them via Dempster-Shafer theory. By explicitly modeling "epistemic conflict" between experts, we enable dynamic uncertainty quantification to prevent overconfident failures on unseen attacks.

**Evidential Opinion Generation.** Standard Softmax classifiers produce point estimates that are unable to distinguish uncertainty arising from lack of evidence versus conflicting evidence. To address this, we adopt Subjective Logic (SL) (Jøsang, 2018) to map features into Dirichlet-parameterized subjective opinions. For each view index $v \in \{1, 2\}$, where $v = 1$ corresponds to the semantic view feature $f_s$ and $v = 2$ to the artifact view feature $f_a$. We project the feature $f_v$ into a non-negative evidence vector $\mathbf{e}^v \in \mathbb{R}^K$ through a dedicated evidential head. This head is implemented as a lightweight multi-layer perceptron (MLP) followed by a Softplus activation to ensure non-negativity:

$$e^v = \text{Softplus}(W^v f_v + C^v), \tag{5}$$

where $e_k^v \geq 0$ represents the observed support for class $k$ (with K=2 corresponding to Real and Fake). The resulting evidence is associated with a Dirichlet distribution parameterized by $\alpha_k^v = e_k^v + 1$, from which a subjective opinion $T^v = \{b^v, u^v\}$ is derived, consisting of belief masses $b^v$ and epistemic uncertainty $u^v$. Letting $S^v = \sum_k \alpha_k^v$ denote the Dirichlet Strength, we compute:

$$b_k^v = \frac{e_k^v}{S^v}, \quad u^v = \frac{K}{S^v}, \tag{6}$$

subject to the additivity constraint $\sum_k b_k^v + u^v = 1$.

This formulation yields two decorrelated evidential experts: (1) **Content Expert ($T^1$):** derived from $f_s$, reflecting beliefs based on high-level identity and semantic consistency. (2) **Structure Expert ($T^2$):** derived from $f_a$, reflecting beliefs based on low-level manifold anomalies. Crucially, this evidential representation naturally captures ignorance. When a view lacks discriminative cues ($e_k \approx 0$), the total strength $S^v$ decreases, causing the uncertainty $u^v \to 1$. This mechanism prevents blind overconfident predictions and provides a principled foundation for subsequent conflict-aware fusion.

**Conflict-Aware Evidential Fusion.** To exploit the complementarity of the dual-view experts, we employ the Dempster-Shafer (DS) combination rule to produce a robust global decision. Unlike traditional averaging, which obscures potential risks by smoothing discrepancies, DS theory (Dempster, 1968) explicitly models inter-view consistency and conflict through orthogonal sum $T = T^1 \oplus T^2$. For class $k \in \{1, 2\}$, the fused belief mass $b_k$ and uncertainty $u$ are computed as:

$$b_k = \frac{1}{1 - \mathcal{C}} \left( b_k^1 b_k^2 + b_k^1 u^2 + b_k^2 u^1 \right), \quad u = \frac{1}{1 - \mathcal{C}} \left( u^1 u^2 \right), \tag{7}$$

where $\mathcal{C} = \sum_{i \neq j} b_i^1 b_j^2$ denotes the conflict coefficient, quantifying the degree of inconsistency between the semantic and artifact views. The normalization factor $(1 - \mathcal{C})^{-1}$ redistributes belief mass associated with conflicting evidence, ensuring $\sum b_k + u = 1$.

**Why is this mechanism effective?** The fusion strategy goes beyond numerical weighting by embedding a logic that explicitly handles three critical forensic scenarios: (1) **Consensus Amplification:** When both experts agree with high confidence, the term $b_k^1 b_k^2$ dominates, yielding a synergistic increase in fused belief, while $u$ decays rapidly ($u \approx u^1 u^2$), resulting in substantially strengthened confidence. (2) **Knowledge Complementarity:** When one view is confident (high $b_k^1$, low $u^1$) and the other is uncertain (high $u^2$), the cross-term $b_k^1 u^2$ allows the confident view to compensate for the ignorance of the other. This enables unilateral knowledge filling, preventing missed detections by single-view failure. (3) **Conflict Awareness:** When views contradict (e.g., Semantic="Real" vs. Artifact="Fake"), the coefficient $\mathcal{C}$ rises significantly. Instead of forcing a blind deterministic decision, the normalization suppresses belief mass and elevates epistemic uncertainty, producing a reliable risk signal for rejecting ambiguous samples.

**Inference.** During testing, class probabilities are obtained via the Dirichlet mean $p_k = b_k + u/K$, while the fused uncertainty $u$ serving as the epistemic confidence score.

### 3.5. Loss Function

To effectively optimize the proposed "Divide-and-Conquer" framework, we design a composite objective function that governs both feature decoupling and evidential reasoning. Specifically, the optimization is driven by two complementary constraints: (1) **Semantic Manifold Loss** supervises the manifold projector in the "Divide" phase to ensure precise view construction; (2) **Uncertainty-Aware Evidential Loss** guides the opinion generation in the "Conquer" phase to ensure reliable decision-making.

**Semantic Manifold Loss.** To reconstruct a continuous semantic manifold, we employ a $\beta$-VAE as the manifold projector to approximate the generative distribution of CLIP features ($f_s \to f_c$). Since CLIP embeddings are inherently hyperspherical and encode semantics primarily in angular directions rather than vector magnitudes, conventional $\ell_2$ reconstruction objectives are suboptimal and may overfit non-semantic scale variations. We therefore adopt a cosine distance-based alignment loss, explicitly enforcing directional consistency between the original and reconstructed features. The manifold loss $\mathcal{L}_{vae}$ is formulated as:

$$\mathcal{L}_{vae} = \mathcal{L}_{align} + \beta \cdot \mathcal{L}_{kld}, \tag{8}$$

To enforce directional consistency, we minimize the cosine distance between the original semantic features $f_s$ and the reconstructed features $f_c$:

$$\mathcal{L}_{align} = 1 - \cos(f_s, f_c) = 1 - \frac{\langle f_s, f_c \rangle}{\|f_s\|_2 \|f_c\|_2}, \tag{9}$$

This constraint compels the manifold projector to prioritize semantic orientation over numerical magnitude. To ensure

Table 1. Quantitative comparison on cross-dataset and cross-manipulation benchmarks (best in bold).

| Methods | Venue | Cross-dataset Evaluation | | | | | | | Cross-manipulation Evaluation | | | | | | | | |
|---------|-------|-------|------|------|------|------|-------|------|--------|---------|---------|------|---------|------|--------|---------|------|
| | | CDFv2 | DFD | DFDC | DFo | WDF | CDFv3 | Avg. | UniFace | BleFace | MobSwap | e4s | FaceDan | FSGAN | InSwap | SimSwap | Avg. |
| F3Net | ECCV'20 | 0.789 | 0.844 | 0.718 | 0.730 | 0.728 | 0.736 | 0.758 | 0.809 | 0.808 | 0.867 | 0.494 | 0.717 | 0.845 | 0.757 | 0.674 | 0.746 |
| SPSL | CVPR'20 | 0.799 | 0.871 | 0.724 | 0.723 | 0.702 | 0.740 | 0.760 | 0.747 | 0.748 | 0.885 | 0.514 | 0.666 | 0.812 | 0.643 | 0.665 | 0.710 |
| SRM | CVPR'21 | 0.840 | 0.885 | 0.695 | 0.722 | 0.702 | 0.793 | 0.773 | 0.749 | 0.704 | 0.779 | 0.704 | 0.659 | 0.772 | 0.793 | 0.694 | 0.732 |
| CORE | CVPR'21 | 0.809 | 0.882 | 0.721 | 0.765 | 0.724 | 0.750 | 0.775 | 0.871 | 0.843 | 0.959 | 0.679 | 0.774 | 0.958 | 0.855 | 0.724 | 0.833 |
| RECCE | CVPR'22 | 0.823 | 0.891 | 0.696 | 0.784 | 0.756 | 0.809 | 0.793 | 0.898 | 0.832 | 0.925 | 0.683 | 0.848 | 0.949 | 0.848 | 0.768 | 0.844 |
| SBI | CVPR'22 | 0.886 | 0.827 | 0.717 | 0.899 | 0.703 | 0.738 | 0.795 | 0.724 | 0.891 | 0.952 | 0.750 | 0.594 | 0.803 | 0.712 | 0.701 | 0.766 |
| UCF | ICCV'23 | 0.837 | 0.867 | 0.742 | 0.808 | 0.774 | 0.761 | 0.798 | 0.831 | 0.827 | 0.950 | 0.731 | 0.862 | 0.937 | 0.809 | 0.647 | 0.824 |
| IID | CVPR'23 | 0.838 | 0.939 | 0.700 | 0.808 | 0.666 | 0.760 | 0.785 | 0.839 | 0.789 | 0.888 | 0.766 | 0.844 | 0.927 | 0.789 | 0.644 | 0.811 |
| LSDA | CVPR'24 | 0.875 | 0.881 | 0.701 | 0.768 | 0.797 | 0.727 | 0.792 | 0.872 | 0.875 | 0.930 | 0.694 | 0.721 | 0.939 | 0.855 | 0.793 | 0.835 |
| ProDet | NIPS'24 | 0.926 | 0.901 | 0.707 | 0.879 | 0.781 | 0.732 | 0.821 | 0.908 | 0.929 | 0.975 | 0.771 | 0.747 | 0.928 | 0.837 | 0.844 | 0.867 |
| Effort | ICML'25 | 0.956 | 0.965 | 0.843 | 0.977 | 0.848 | 0.850 | 0.907 | 0.962 | 0.873 | 0.953 | 0.983 | 0.926 | 0.957 | 0.936 | 0.926 | 0.940 |
| GenD | WACV'26 | 0.960 | 0.970 | 0.871 | 0.989 | 0.890 | 0.859 | 0.923 | 0.974 | 0.935 | 0.976 | 0.983 | 0.942 | 0.974 | 0.943 | 0.938 | 0.958 |
| **Ours** | – | **0.977** | **0.982** | **0.882** | **0.993** | **0.911** | **0.886** | **0.939** | **0.982** | **0.963** | **0.978** | **0.991** | **0.972** | **0.989** | **0.970** | **0.965** | **0.976** |

manifold smoothness, we minimize the KL divergence between the posterior distribution and a standard normal prior:

$$
\begin{aligned}
\mathcal{L}_{kld} &= D_{KL}(q(z|f_s)\|\mathcal{N}(0,I)) \\
&= -\frac{1}{2}\sum(1+\log\sigma^2-\mu^2-\sigma^2),
\end{aligned}
\tag{10}
$$

where $\mu$ and $\sigma^2$ denote the mean and variance of the output predicted by the encoder.

**Uncertainty-Aware Evidential Loss.** We leverage Evidential Deep Learning (Sensoy et al., 2018) to equip the model with uncertainty quantification capabilities. Instead of predicting point estimates, the network predicts the concentration parameters $\boldsymbol{\alpha}_i$ of a Dirichlet distribution (Ng et al., 2011) $D(\mathbf{p}_i|\boldsymbol{\alpha}_i)$ over the $K$-dimensional probability simplex:

$$
D(\mathbf{p}_i|\alpha_i) = \begin{cases} \frac{1}{B(\alpha_i)}\prod_{k=1}^{K}p_{i,k}^{\alpha_{i,k}-1} & \text{for } \mathbf{p}_i \in \mathcal{S}_K, \\ 0 & \text{otherwise,} \end{cases}
\tag{11}
$$

To optimize the model parameters, we minimize the *Bayes risk* with respect to the cross-entropy loss. This objective encourages the network to accumulate evidence for the ground-truth class:

$$
\mathcal{L}_{fit}(\alpha_i) = \int \left[\sum_{k=1}^{K}-y_{i,k}\log(p_{i,k})\right]\frac{1}{B(\alpha_i)}\prod_{k=1}^{K}p_{i,k}^{\alpha_{i,k}-1}d\mathbf{p}_i,
\tag{12}
$$

Analytically, this integral simplifies to a tractable form using the Digamma function $\psi(\cdot)$:

$$
\mathcal{L}_{fit}(\alpha_i) = \sum_{k=1}^{K}y_{i,k}(\psi(S_i)-\psi(\alpha_{i,k})),
\tag{13}
$$

where $S_i = \sum_{k=1}^{K}\alpha_{i,k}$ represents the total Dirichlet strength.

To suppress misleading evidence for incorrect classes, we incorporate a KL divergence regularization term that penalizes

deviations from a uniform prior for non-target classes:

$$
KL[D(\mathbf{p}_i \mid \widetilde{\alpha}_i)\|D(\mathbf{p}_i \mid \mathbf{1})] = \log\left(\frac{\Gamma(\sum_{k=1}^{K}\tilde{\alpha}_{i,k})}{\Gamma(K)\prod_{k=1}^{K}\Gamma(\tilde{\alpha}_{i,k})}\right)
$$
$$
+ \sum_{k=1}^{K}(\tilde{\alpha}_{i,k}-1)\left[\psi(\tilde{\alpha}_{i,k})-\psi\left(\sum_{j=1}^{K}\tilde{\alpha}_{i,j}\right)\right],
\tag{14}
$$

Here, $\widetilde{\boldsymbol{\alpha}}_i = \mathbf{y}_i + (1-\mathbf{y}_i)\odot\boldsymbol{\alpha}_i$ is the masked parameter vector, which ensures that evidence for the ground-truth class is not penalized. The final evidential loss is defined as:

$$
\mathcal{L}_{edl} = \frac{1}{N}\sum_{i=1}^{N}(\mathcal{L}_{fit}(\alpha_i)+\lambda_t\cdot KL[D(\mathbf{p}_i \mid \widetilde{\alpha}_i)\|D(\mathbf{p}_i \mid \mathbf{1})]),
\tag{15}
$$

where $\lambda_t = \min(1,t/T) \in [0,1]$ is an annealing coefficient that gradually increases the regularization weight over training epochs $t$.

**Total Objective.** The complete training objective integrates evidential learning with semantic manifold regularization:

$$
\mathcal{L}_{total} = \mathcal{L}_{edl} + \lambda_{vae}\mathcal{L}_{vae},
\tag{16}
$$

where $\lambda_{vae}$ balances discriminative supervision against semantic structure preservation.

## 4. Experiments

### 4.1. Experimental Settings

**Implementation Details.** We adopt CLIP ViT-L/14 (Radford et al., 2021) as the backbone and follow the standard preprocessing pipeline of DeepfakeBench (Yan et al., 2023b). All experiments are conducted on a single NVIDIA A100 GPU with a batch size of 128. The model is optimized using AdamW (Loshchilov & Hutter, 2017) with an initial learning rate of $1\times10^{-4}$. To improve robustness, we apply common data augmentations, including Gaussian blur, JPEG compression, and color jittering. Performance is reported using video-level AUC.

**Evaluation Protocols and Datasets.** For all experiments, the model is trained on FaceForensics++ (FF++) (Rossler et al., 2019) under the c23 compression setting. We evaluate performance under two standard protocols: **(1) Cross-dataset evaluation**, testing on six widely used benchmarks (CDFv2 (Li et al., 2020b), DFD (Google AI, 2020), DFDC (Deepfake Detection Challenge, 2020), DFo (Jiang et al., 2020), WDF (Zi et al., 2020), and CDFv3 (Li et al., 2025)) to assess domain generalization; and **(2) Cross-manipulation evaluation**, conducted on DF40 (Yan et al., 2024b), which shares the FF++ source domain but contains 40 distinct manipulation types, isolating robustness to forgery diversity from domain shift.

## 4.2. Comparisons with Existing Methods

To establish a comprehensive benchmark, we compare our approach against 12 representative methods, ranging from classic baselines to the latest advances. These include F3Net (Qian et al., 2020), SPSL (Liu et al., 2021), SRM (Luo et al., 2021), CORE (Ni et al., 2022), RECCE (Cao et al., 2022), SBI (Shiohara & Yamasaki, 2022), UCF (Yan et al., 2023a), IID (Huang et al., 2023), LSDA (Yan et al., 2024a), ProDet (Cheng et al., 2024), Effort (Yan et al., 2025), and GenD (Yermakov et al., 2026) . As shown in Table 1, our method outperforms all baselines in the challenging 'unseen domains + attacks' setting, achieving the best performance across all benchmarks. Furthermore, even when domain gaps are minimized using DF40, our approach maintains superior detection rates. This confirms that our model captures universal manifold anomalies rather than overfitting specific patterns, ensuring robust generalization against both unseen domains and unknown attacks.

## 4.3. Visualization and Interpretability

**Uncertainty Estimation and Risk Rejection.** To validate our uncertainty estimation, we analyze the epistemic uncertainty $u$ in Figure 3 derived via EDL across FF++ and cross-domain benchmarks (CDFv2, DFDC, DFD). First, distribution analysis reveals a decisive separation: correct predictions are tightly clustered near $u \rightarrow 0$, whereas misclassifications predominantly populate the high-uncertainty spectrum. This confirms that the model successfully aligns high uncertainty with prediction error, effectively identifying its own knowledge boundaries. Second, risk-coverage analysis demonstrates a strictly monotonic increase in accuracy as the retention ratio decreases. Notably, rejecting just the top 10% of uncertain samples on DFD yields substantial performance gains. This validates $u$ as a robust indicator for risk-aware deployment.

**Multi-view Complementarity.** To verify that the fusion strategy captures heterogeneous and complementary feature representations, we visualize attention maps across views in Figure 4. The Semantic View focuses on global facial

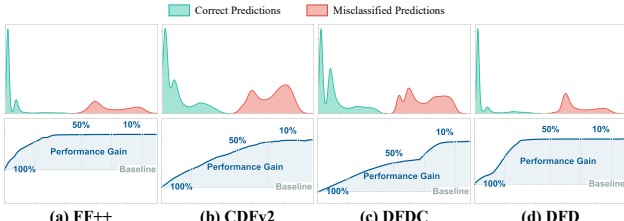

*Figure 3.* Evaluation of Uncertainty Effectiveness. Top: Uncertainty distributions clearly separate correct and misclassified predictions, with errors concentrated at high uncertainty. Bottom: Risk–coverage curves show monotonic accuracy gains when rejecting uncertain samples, consistently outperforming the baseline.

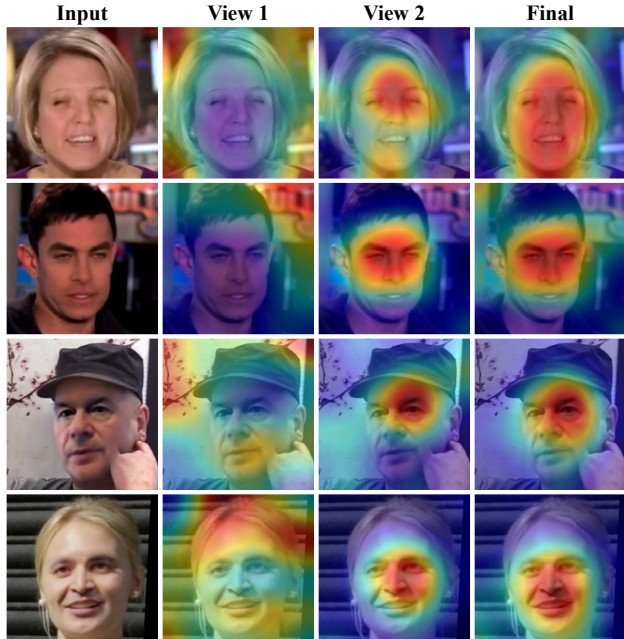

*Figure 4.* Grad-CAM visualization. View 1 focuses on global semantics, while View 2 targets local artifacts. The final fusion effectively synergizes these complementary cues, which corrects single-view biases to achieve precise localization.

structures, consistent with CLIP's high-level semantic priors. Conversely, the Artifact View specifically targets local anomalies. Crucially, the Final View is not a naive superposition but a dynamic synthesis driven by adaptive evidential weighting. By integrating global semantics with local manifold anomalies, our model guarantees robust and precise decision-making.

## 4.4. Ablation Study and Analysis

**Analysis of Architectural Mechanisms.** Table 2(a) provides an in-depth analysis of the intrinsic mechanisms underlying our multi-view architecture and feature disentanglement strategy. First, regarding the necessity of the Artifact View (A), the single-stream baseline relying solely on CLIP features ($f_s$) suffers a significant performance drop, which confirms that pure semantics are insufficient for detecting high-fidelity deepfakes. This is visually corroborated by the

*Table 2.* Comprehensive ablation studies evaluated on CDFv2, DFDC, and MobileFaceSwap (MFS) from the CDFv3 dataset.

*(a)* Architectural mechanisms.

| Variant | View Comp. | Ortho. | CDFv2 | DFDC | MFS | Avg. |
|---------|-----------|--------|-------|------|-----|------|
| (A) | $f_s$ | - | 0.927 | 0.856 | 0.933 | 0.905 |
| (B) | $f_s + f_s$ | - | 0.954 | 0.867 | 0.929 | 0.917 |
| (C) | $f_s + f_r$ | × | 0.956 | 0.860 | 0.951 | 0.922 |
| **Ours** | $f_s + f_a$ | ✓ | **0.977** | **0.882** | **0.956** | **0.938** |

*(b)* Decision logic and fusion strategy.

| Variant | Loss | Fusion | Unc. | CDFv2 | DFDC | MFS | Avg. |
|---------|------|--------|------|-------|------|-----|------|
| (D) | $\mathcal{L}_{ce}$ | Mean | × | 0.956 | 0.874 | 0.942 | 0.924 |
| (E) | $\mathcal{L}_{edl}$ | Mean | ✓ | 0.961 | 0.878 | 0.941 | 0.927 |
| **Ours** | $\mathcal{L}_{edl}$ | DS-C. | ✓ | **0.977** | **0.882** | **0.956** | **0.938** |

*(c)* Alignment and KL-divergence losses.

| Variant | $\mathcal{L}_{align}$ | $\mathcal{L}_{kld}$ | CDFv2 | DFDC | MFS | Avg. |
|---------|------|------|-------|------|-----|------|
| (F) | × | ✓ | 0.955 | 0.866 | 0.944 | 0.922 |
| (G) | ✓ | × | 0.948 | 0.872 | 0.944 | 0.921 |
| **Ours** | ✓ | ✓ | **0.977** | **0.882** | **0.956** | **0.938** |

*(d)* Backbones and pre-training.

| Backbone | Pre-train | Patch | CDFv2 | DFDC | MFS | Avg. |
|----------|-----------|-------|-------|------|-----|------|
| CLIP-B (H) | Text-Img | 16 | 0.886 | 0.811 | 0.790 | 0.829 |
| DINO-L (I) | Self-Sup. | 14 | 0.856 | 0.796 | 0.773 | 0.808 |
| **CLIP-L** | **Text-Img** | **14** | **0.977** | **0.882** | **0.956** | **0.938** |

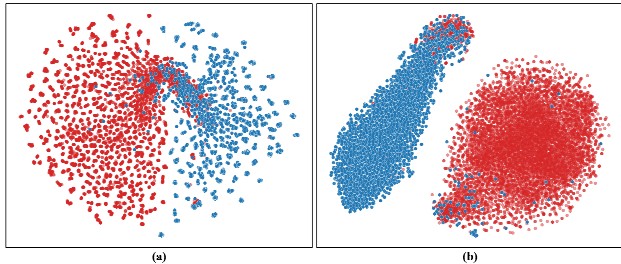

*Figure 5.* T-SNE Visualizations. (a) Baseline: Relying solely on CLIP semantics results in severe mixing of real and fake samples, indicating failure to distinguish deepfakes. (b) Ours: By employing Geometric View Purification, our method effectively isolates artifacts from semantics. The resulting distributions form compact clusters separated by a wide decision margin, validating the discriminative power of the purified artifact view.

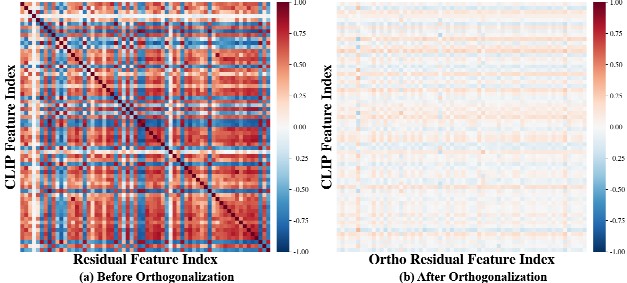

*Figure 6.* Feature Correlation Analysis. (a) Pronounced correlations between CLIP semantic features and raw residuals reveal severe semantic leakage in a strongly coupled regime. (b) The sparse heatmap confirms that our geometric projection substantially weakens the shared information pathway, achieving an approximately decorrelated, complementary state for reliable fusion.

severe entanglement in the t-SNE plot (Figure 5(a). Second, to exclude ensemble effects (B), the inferiority of the homogeneous dual-stream variant ($f_s + f_s$) confirms that performance gains stem from feature heterogeneity rather than increased model capacity or network depth. Finally, regarding the importance of Orthogonal Disentanglement (C), removing the geometric constraint leads to generalization degradation. We attribute this to semantic leakage between CLIP semantics and original residual features, which is explicitly visualized in Figure 6(a) as strong and structured cross-feature correlations that violate the requirement for

view decorrelation and complementarity. In contrast, our orthogonal projection effectively enforces feature decoupling, yielding a sparse, structure-free cross-correlation heatmap in Figure 6(b). This disentanglement is also observed in Figure 5(b), where distinct and compact clusters emerge, separated by a wide and well-defined decision margin, indicating that pure and discriminative forensic traces are successfully isolated from semantic content.

*Table 3.* Comparison of Expected Calibration Error (ECE) across different datasets. Lower is better.

| Method | CDFv2 | DFD | MFS | Avg. |
|--------|-------|-----|-----|------|
| Baseline (D) | 0.13 | 0.15 | 0.19 | 0.16 |
| **Ours** | **0.07** | **0.07** | **0.08** | **0.07** |

**Analysis of Decision Logic and Fusion Strategy.** We further analyze the impact of probabilistic modeling and fusion rules in Table 2(b). First, regarding mitigating overconfidence (D), we replace the proposed method with standard Cross-Entropy Loss ($\mathcal{L}_{ce}$), forcing the model into a deterministic point estimation regime. The performance drop confirms that deterministic methods suffer from severe overconfidence on unseen domains. As shown in Table 3, our method achieves substantially better calibration for more trustworthy predictions. Second, for handling "epistemic conflict" (E), while retaining the Dirichlet-based uncertainty modeling, we employ Naive Mean Fusion to integrate the two views by simply averaging their belief masses. The results prove that simple linear superposition fails to resolve inter-view contradictions, whereas Dempster-Shafer rule explicitly manages "epistemic conflict", ensuring robustness against ambiguous samples.

**Analysis of Manifold Constraints.** Table 2(c) further validates the necessity of internal constraints within the manifold view, relying on the synergy of Semantic Alignment Loss ($\mathcal{L}_{align}$) and KL Divergence ($\mathcal{L}_{kld}$). First, regarding the necessity of semantic alignment (G), we remove $\mathcal{L}_{align}$, forcing the model to optimize reconstruction relying solely on the standard MSE loss. The resulting performance drop confirms that Euclidean optimization is incompatible with CLIP's hyperspherical geometry, inevitably introducing se-

mantic bias into the residuals. Second, for the necessity of latent regularization (F), we exclude the KL divergence constraint, degenerating the $\beta$-VAE into a standard deterministic Autoencoder. The observed degradation verifies that regularization is vital to prevent rote memorization, ensuring the model learns a generalized normal distribution for robust detection.

**Analysis of Backbone Adaptability.** Table 2(d) examines backbone adaptability. First, regarding scale (H), the performance drop with ViT-B/16 confirms that larger capacity (ViT-L/14) is indispensable for resolving subtle artifacts. Second, regarding priors (I), the inferiority of DINOv2 underscores the value of CLIP. Unlike pure visual features, CLIP's text-image alignment establishes a stable "semantic anchor", which is critical for defining manifold anomalies.

## 5. Conclusion

In this paper, we tackle the challenge of trustworthy deepfake detection by introducing a novel Divide-and-Conquer framework termed Reliable Multi-View Evidential Learning. To mitigate the *Semantic Masking Effect*, we propose Geometric View Purification which projects residual representations onto the orthogonal complement of the semantic space, thereby explicitly disentangling forensic artifacts from semantic interference. Building upon this decoupling, we further develop Uncertainty-Aware Evidential Learning, leveraging Dempster–Shafer theory to model "epistemic conflict" and quantify predictive uncertainty in a principled manner. Extensive experiments across diverse benchmarks demonstrate that our framework not only improves cross-domain generalization but also yields reliable uncertainty estimates, highlighting its potential for deployment in safety-critical and open-world deepfake detection scenarios.

## Acknowledgements

This work was supported by the Key Science and Technology Research Project of Xinjiang Production and Construction Corps under Grant No.2025AB029, and the National Natural Science Foundation of China under Grants No.62371350, No.62501248, and No.U2441240.

## Impact Statement

This work aims to mitigate deepfake threats and restore digital trust, protecting against identity theft and disinformation. While acknowledging the dual-use risk where detectors could facilitate adversarial generation, we believe the defensive value of open-sourcing outweighs potential misuse. Furthermore, we emphasize the importance of fairness; real-world deployment requires continuous monitoring to ensure the model remains unbiased across demographics.

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

# A. Appendix

This appendix provides detailed implementation settings, algorithmic procedures, and extensive experimental analysis to support the main paper. The content is organized as follows:

- **Implementation Details:** Detailed hyperparameters and optimization settings (see Sec. A.1).
- **Training Algorithm:** The complete pseudocode for the proposed framework (see Sec. A.2).
- **Extensive Experimental Analysis:** A comprehensive evaluation including a comparative analysis of model efficiency (see Sec. A.3).
- **Limitations and Future Work:** A discussion on the current limitations, potential future directions, and ethical considerations (see Sec. A.4).

## A.1. Implementation Details

Our method is implemented in PyTorch on a single NVIDIA A100 GPU. We adopt the CLIP ViT-L/14 as the frozen backbone and apply LoRA ($r = 8$) to the attention layers for parameter-efficient adaptation. The model is optimized using AdamW with cosine learning rate decay. For the loss balancing, we set $\lambda_{vae} = 0.7$ and $\beta = 2.0$ to regulate the manifold learning. Detailed hyperparameter settings are listed in Table 4.

*Table 4.* Hyperparameter Settings.

| Category | Hyperparameter | Value |
|---|---|---|
| **Optimization** | Input Resolution | $224 \times 224$ |
| | Batch Size | 128 |
| | Optimizer | AdamW |
| | Betas | (0.9, 0.999) |
| | Weight Decay | $1 \times 10^{-2}$ |
| | Learning Rate | $1 \times 10^{-4}$ |
| | Total Epochs | 20 |
| **Model & PEFT** | Backbone | CLIP ViT-L/14 |
| | LoRA Rank ($r$) | 8 |
| | LoRA Alpha ($\alpha$) | 16 |
| | LoRA Dropout | 0.1 |
| **Loss Coefficients** | KL Weight ($\beta$) | 2.0 |
| | Global VAE Weight ($\lambda_{vae}$) | 0.7 |

## A.2. Training Algorithm

We provide the detailed training procedure of our proposed framework in **Algorithm 1**.

## A.3. Extensive experimental analysis

**Complexity Analysis.** We present the comparison of trainable parameters in Table 5 and a supplementary efficiency profiling including GFLOPs, inference time, and FPS in Table 6. We notice that recent parameter-efficient works

---

**Algorithm 1** Reliable Multi-View Evidential Learning Training Algorithm

**Require:** Training data $\mathcal{D} = \{(x_i, y_i)\}_{i=1}^{N}$; Pre-trained CLIP encoder $\mathcal{E}_{clip}$; Hyperparameters $\lambda_{vae}, \beta$.
**Ensure:** Optimized model parameters $\theta_{lora}, \theta_{vae}, \theta_{head}$.
1: **Initialization:**
2: Freeze CLIP backbone parameters $\theta_{frozen}$.
3: Initialize LoRA parameters $\theta_{lora}$, $\beta$-VAE projector $\theta_{vae}$, and evidential heads $\theta_{head}$.
4: **Training Loop:**
5: **for** each epoch **do**
6:   **for** each batch $(x, y)$ in $\mathcal{D}$ **do**
7:     ▷ *Phase 1: Divide (Multi-View Construction)*
8:     **1. Semantic View Extraction:**
9:     $f_s = \mathcal{E}_{clip}(x; \theta_{frozen}, \theta_{lora})$
10:     **2. Manifold Projection:**
11:     $f_c, \mu, \sigma = \beta\text{-VAE}(f_s; \theta_{vae})$
12:     Compute cosine alignment: $\mathcal{L}_{align} = 1 - \frac{\langle f_s, f_c \rangle}{\|f_s\|_2 \|f_c\|_2}$
13:     $\mathcal{L}_{vae} = \mathcal{L}_{align} + \beta \cdot D_{KL}(q(z|f_s)\|\mathcal{N}(0, I))$
14:     **3. Geometric View Purification:**
15:     Compute raw residual: $f_r = f_s - f_c$
16:     Compute semantic leakage: $f_r^{\|} = \frac{\langle f_r, f_s \rangle}{\|f_s\|_2^2} f_s$
17:     Obtain purified artifact view: $f_a = f_r - f_r^{\|}$
18:     ▷ *Phase 2: Conquer (Evidential Learning)*
19:     **4. Opinion Generation:**
20:     Compute evidence for each view $v \in \{1, 2\}$:
21:     $e^v = \text{Softplus}(W^v f_v + C^v)$
22:     Derive Dirichlet parameters: $\alpha_k^v = e_k^v + 1$
23:     Calculate Beliefs $b^v$ and Uncertainty $u^v$
24:     **5. Conflict-Aware Fusion:**
25:     Fuse opinions via Dempster-Shafer rule:
26:     $b_{fused}, u_{fused} = \text{DS\_Combine}(b^1, u^1, b^2, u^2)$
27:     ▷ *Optimization*
28:     **6. Loss Calculation & Update:**
29:     Compute EDL loss $\mathcal{L}_{edl}$ for views 1, 2 and fused output.
30:     Total Loss: $\mathcal{L}_{total} = \mathcal{L}_{edl} + \lambda_{vae}\mathcal{L}_{vae}$
31:     Update $\theta_{lora}, \theta_{vae}, \theta_{head}$ via gradient descent.
32:   **end for**
33: **end for**
34: **return** Trained model parameters.

---

like GenD (Yermakov et al., 2026) and Effort (Yan et al., 2025) achieve remarkable efficiency by tuning LayerNorm parameters (0.104 M) or adapting SVD residuals (0.199 M), respectively. While effective, these methods primarily rely on feature rescaling or linear weight decomposition.

In contrast, our method (0.868 M) employs LoRA to introduce learnable transformation matrices. This additional capacity is architecturally necessary for our Geometric View Purification module, which requires explicitly rotating the residual features to the orthogonal complement of the semantic space to construct an artifact view. Although this geometric disentanglement incurs a slight increase in parameters compared to GenD and Effort, Table 6 demonstrates that our model remains highly efficient during inference, operating at 47.37 FPS with 156.08 GFLOPs. Ultimately, we argue that 0.868 M represents an optimal 'sweet spot':

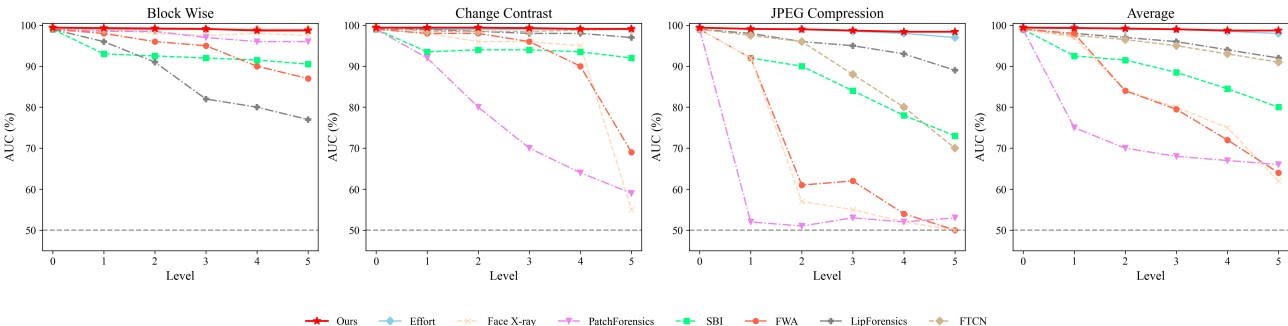

*Figure 7.* Robustness evaluation against common image perturbations. We compare the AUC performance of our method against state-of-the-art baselines under three types of distortions: Block-wise distortion, Contrast change, and JPEG compression, across five severity levels. The rightmost plot shows the average performance. Our method (red star) demonstrates superior stability, maintaining near-perfect detection performance even under severe degradations, significantly outperforming other forensic methods.

it is substantial enough to support complex geometric disentanglement that simple rescaling cannot achieve, while remaining over 99% more parameter-efficient than traditional full fine-tuning methods (e.g., 133 M for LSDA).

*Table 5.* Comparison of Trainable Parameters. Our method achieves a superior trade-off between efficiency and performance.

| Method | Backbone | Trainable Param. |
|---|---|---|
| *Full Fine-tuning Methods* | | |
| F3Net | Xception | 22 M |
| SPSL | Xception | 21 M |
| SRM | Xception | 55 M |
| RECCE | Xception | 48 M |
| IID | ResNet18 | 66 M |
| ProDet | EFNB4 | 96 M |
| LSDA | EFNB4 | 133 M |
| *Parameter-Efficient Methods* | | |
| Effort | CLIP ViT-L/14 | 0.199 M |
| GenD | CLIP ViT-L/14 | **0.104 M** |
| Ours | CLIP ViT-L/14 | 0.868 M |

*Table 6.* Efficiency comparison under a unified local profiling protocol. Our method maintains highly competitive inference speed, ensuring practical deployment efficiency.

| Method | GFLOPs | Time (ms/f) | FPS |
|---|---|---|---|
| Effort | 103.893 | 22.71 | 44.04 |
| GenD | 155.634 | 17.98 | 55.61 |
| **Ours** | **156.080** | **21.11** | **47.37** |

**Robust Analysis.** Real-world deepfake detection often faces low-quality inputs due to transmission compression or environmental noise. To evaluate the robustness of our model against these challenges, we benchmark it against seven representative state-of-the-art methods, including Effort (Yan et al., 2025), Face X-ray (Li et al., 2020a), Patch-Forensics (Chai et al., 2020), SBI (Shiohara & Yamasaki,

2022), FWA (Li & Lyu, 2019), LipForensics (Haliassos et al., 2021), and FTCN (Zheng et al., 2021). We conducted experiments under three common perturbations: Block-wise distortion, Contrast change, and JPEG compression, with severity levels ranging from 1 to 5. As illustrated in Figure 7, most existing methods (e.g., PatchForensics, Face X-ray, and SBI) suffer from significant performance degradation as the perturbation level increases. For instance, in the challenging JPEG Compression scenario, pixel-dependent methods drop drastically because high-frequency artifact traces are destroyed by compression algorithms. In contrast, our method (red line) exhibits remarkable stability, maintaining an AUC above 99% across all severity levels. Even compared to the recent SOTA method Effort (Yan et al., 2025), our approach shows superior resilience, particularly in high-level degradations. This validates that our Geometric View Purification, anchored by robust semantic priors, successfully extracts stable forensic features that are invariant to low-level noise.

### A.4. Limitation and Future Work

While the proposed framework demonstrates strong generalization and reliable uncertainty estimation across diverse benchmarks, several directions remain open for further exploration.

First, our formulation adopts CLIP as a semantic anchor to define the reference semantic space. This choice reflects a deliberate design trade-off between generality and interpretability. Investigating adaptive or jointly optimized semantic manifolds may further enhance the expressiveness of the purified artifact view.

Second, Geometric View Purification enforces global orthogonality between semantic and artifact representations. Although this constraint effectively suppresses semantic masking in practice, future work could explore finer-grained or region-adaptive projections to better accommodate in-

creasingly sophisticated forgeries.

Finally, the current study focuses on image-level evidence aggregation. Extending the proposed uncertainty-aware multi-view evidential framework to temporal, cross-modal, or continual learning scenarios is a natural and promising direction toward large-scale, real-world deployment.

**Ethics & Reproducibility.** All of the facial images that are utilized are sourced from publicly available datasets and are accompanied by appropriate citations. This guarantees that there is no infringement upon personal privacy. Codes and checkpoints are available at https://github.com/kxl0825/DiCoME.git.

