# Divide and Conquer: Reliable Multi-View Evidential Learning for Deepfake Detection

## Abstract

With the evolution of generative models, deepfakes have achieved near-perfect semantic realism, leaving forensic traces only in subtle structural anomalies. However, existing single-view paradigms often fail to generalize, as dominant semantic features overwhelm subtle artifact cues within entangled representations. This imbalance leads to overconfident yet brittle predictions—a phenomenon we term the *Semantic Masking Effect*. To address this challenge, we propose *Reliable Multi-View Evidential Learning for Deepfake Detection* under a "Divide-and-Conquer" strategy. In the "Divide" phase, we employ Geometric View Purification to decompose the entangled representation space through principled geometric projection. This process suppresses semantic interference within artifact-sensitive representations, forming the foundation for independent semantic and artifact views. In the "Conquer" phase, we leverage Uncertainty-Aware Evidential Learning to synthesize these distinct views. By explicitly modeling the "epistemic conflict" between semantic and artifact cues, this mechanism provides calibrated uncertainty estimates instead of forcing rigid deterministic decisions. Extensive experiments across multiple benchmarks demonstrate that our method consistently outperforms existing approaches in generalization performance, while providing reliable uncertainty estimation for trustworthy deepfake detection.

## 1. Introduction

The democratization of high-fidelity synthesis via GANs (Goodfellow et al., 2014) and Diffusion Models (Ho et al., 2020) has elevated deepfakes from a technological curiosity to a serious security threat, enabling large-scale misinfor-

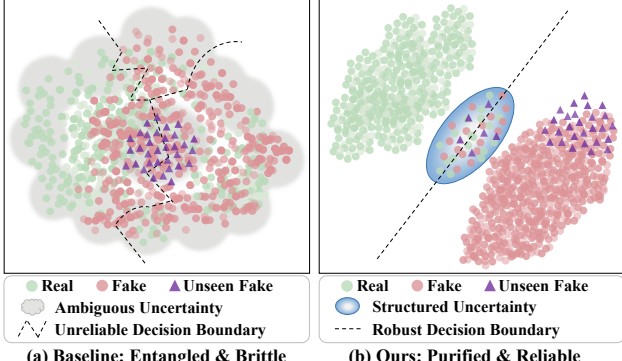

*Figure 1.* Feature entanglement vs. Our solution. (a) Dominant semantic features overshadow subtle artifact cues, causing unseen fake samples to entangle with real data. This leads to ambiguous uncertainty and unreliable decision boundaries that fail on new attacks. (b) Artifact features are isolated from semantic content. Hard unseen samples are captured in a structured uncertainty region, enabling robust boundaries and superior generalization.

mation and financial fraud (Sunil et al., 2025). While generative models continue to advance at a remarkable pace, deepfake detection methods still suffer from a fundamental generalization bottleneck, particularly when facing unseen manipulation techniques and cross-domain scenarios (Lin et al., 2024; Zhang et al., 2025). This gap highlights the urgent need for detection methods that capture universal and transferable forensic cues, rather than overfitting dataset-specific patterns (Chandra et al., 2025).

A fundamental challenge to robust generalization arises from the intrinsic conflict between high-level semantic realism and low-level structural anomalies. Most existing models adopt a single-view holistic processing paradigm (Rossler et al., 2019), employing monolithic backbones to indiscriminately encode both semantic content and forgery traces into an entangled representation. However, visual backbones are typically optimized for semantic invariance, causing high-magnitude identity features to inherently dominate the representation. Consequently, dominant semantic signals tend to overwhelm weak artifact cues within this entangled feature space, a phenomenon we term the *Semantic Masking Effect*. As illustrated in Fig. 1(a), such feature entanglement forces models to learn unreliable decision boundaries that lead to baseless overconfidence on realistic fakes. In contrast, an ideal detector should explicitly

[1]Anonymous Institution, Anonymous City, Anonymous Region, Anonymous Country. Correspondence to: Anonymous Author <anon.email@domain.com>.

Preliminary work. Under review by the International Conference on Machine Learning (ICML). Do not distribute.

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

12:    $\mathcal{L}_{vae} = \|f_s - f_c\|^2 + \beta\mathcal{D}_{KL}(\mathcal{N}(\mu, \sigma)\|\mathcal{N}(0, I))$
13:    **3. Geometric View Purification:**
14:    Compute raw residual: $f_r = f_s - f_c$
15:    Compute semantic leakage: $f_r^{\parallel} = \frac{\langle f_r, f_s \rangle}{\|f_s\|_2^2} f_s$
16:    Obtain purified artifact view: $f_a = f_r - f_r^{\parallel}$
17:    ▷ *Phase 2: Conquer (Evidential Learning)*
18:    **4. Opinion Generation:**
19:    Compute evidence for each view $v \in \{s, a\}$:
20:    $\mathbf{e}^v = \text{Softplus}(\mathbf{W}^v f_v + \mathbf{b}^v)$
21:    Derive Dirichlet parameters: $\boldsymbol{\alpha}^v = \mathbf{e}^v + \mathbf{1}$
22:    Calculate Beliefs $b^v$ and Uncertainty $u^v$
23:    **5. Conflict-Aware Fusion:**
24:    Fuse opinions via Dempster-Shafer rule:
25:    $b_{fused}, u_{fused} = \text{DS\_Combine}(b^s, u^s, b^a, u^a)$
26:    ▷ *Optimization*
27:    **6. Loss Calculation & Update:**
28:    Compute EDL loss $\mathcal{L}_{edl}$ for views $s, a$ and fused output.
29:    Total Loss: $\mathcal{L} = \mathcal{L}_{edl} + \lambda_{vae}\mathcal{L}_{vae}$
30:    Update $\theta_{lora}, \theta_{vae}, \theta_{head}$ via gradient descent.
31:  **end for**
32: **end for**
33: **return** Trained model parameters.

---

by tuning LayerNorm parameters (0.002M) or adapting SVD residuals (0.19M), respectively. While effective, these methods primarily rely on feature rescaling or linear weight decomposition. In contrast, our Geometric View Purification requires constructing a dynamic projection space that explicitly rotates the residual features to the orthogonal complement of the semantics. In contrast, our method (0.87M) employs LoRA to introduce learnable transformation matrices. This additional capacity is architecturally necessary for our Geometric View Purification module, which requires rotating the semantic space to construct an explicit orthogonal artifact view. We argue that 0.87M represents an optimal 'sweet spot': it is substantial enough to support complex geometric disentanglement—which simple rescaling cannot achieve—yet remains over 99% more efficient than traditional full fine-tuning methods.

**Robust Analysis.** Real-world deepfake detection often

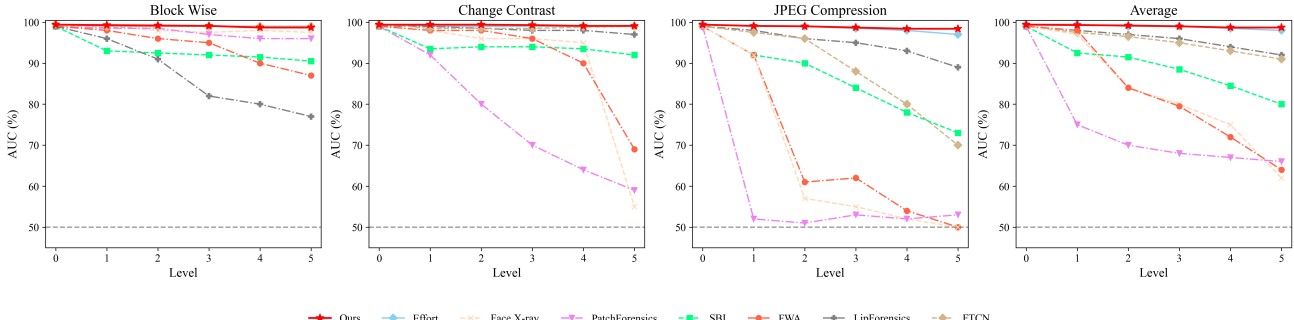

*Figure 7.* Robustness evaluation against common image perturbations. We compare the AUC performance of our method against state-of-the-art baselines under three types of distortions: Block-wise distortion, Contrast change, and JPEG compression, across five severity levels. The rightmost plot shows the average performance. Our method (red star) demonstrates superior stability, maintaining near-perfect detection performance even under severe degradations, significantly outperforming other forensic methods.

*Table 5.* **Comparison of Trainable Parameters.** Our method achieves a superior trade-off between efficiency and performance.

| Method | Backbone | Trainable Param. |
|---|---|---|
| *Full Fine-tuning Methods* | | |
| F3Net | Xception | 22 M |
| SPSL | Xception | 21 M |
| SRM | Xception | 55 M |
| RECCE | Xception | 48 M |
| IID | ResNet18 | 66 M |
| ProDet | EFNB4 | 96 M |
| LSDA | EFNB4 | 133 M |
| *Parameter-Efficient Methods* | | |
| Effort | CLIP ViT-L/14 | 0.19 M |
| GenD | CLIP ViT-L/14 | **0.002 M** |
| **Ours** | **CLIP ViT-L/14** | **0.87 M** |

faces low-quality inputs due to transmission compression or environmental noise. To evaluate the robustness of our model against these challenges, we benchmark it against seven representative state-of-the-art methods, including Effort (Yan et al., 2025), Face X-ray (Li et al., 2020a), Patch-Forensics (Chai et al., 2020), SBI (Shiohara & Yamasaki, 2022), FWA (Li & Lyu, 2019), LipForensics (Haliassos et al., 2021), and FTCN (Zheng et al., 2021). We conducted experiments under three common perturbations: Block-wise distortion, Contrast change, and JPEG compression, with severity levels ranging from 1 to 5. As illustrated in Figure 7, most existing methods (e.g., PatchForensics, Face X-ray, and SBI) suffer from significant performance degradation as the perturbation level increases. For instance, in the challenging JPEG Compression scenario, pixel-dependent methods drop drastically because high-frequency artifact traces are destroyed by compression algorithms. In contrast, our method (red line) exhibits remarkable stability, maintaining an AUC above 99% across all severity levels. Even compared to the recent SOTA method Effort (Yan et al., 2025), our approach shows superior resilience, par-

ticularly in high-level degradations. This validates that our Geometric View Purification, anchored by robust semantic priors, successfully extracts stable forensic features that are invariant to low-level noise.

**A.4. Limitation and Future Work**

While the proposed framework demonstrates strong generalization and reliable uncertainty estimation across diverse benchmarks, several directions remain open for further exploration.

First, our formulation adopts CLIP as a semantic anchor to define the reference semantic space. This choice reflects a deliberate design trade-off between generality and interpretability. Investigating adaptive or jointly optimized semantic manifolds may further enhance the expressiveness of the purified artifact view.

Second, Geometric View Purification enforces global orthogonality between semantic and artifact representations. Although this constraint effectively suppresses semantic masking in practice, future work could explore finer-grained or region-adaptive projections to better accommodate increasingly sophisticated forgeries.

Finally, the current study focuses on image-level evidence aggregation. Extending the proposed uncertainty-aware multi-view evidential framework to temporal, cross-modal, or continual learning scenarios is a natural and promising direction toward large-scale, real-world deployment.

**Ethics & Reproducibility.** All of the facial images that are utilized are sourced from publicly available datasets and are accompanied by appropriate citations. This guarantees that there is no infringement upon personal privacy. We will make all codes and checkpoints available for public access upon acceptance.