# OpenReview forum: "Divide and Conquer: Reliable Multi-View Evidential Learning for Deepfake Detection"
_ICML.cc/2026/Conference — ICML 2026 regular_

### Official Review · Reviewer_sFYT · 2026-03-01

**Soundness:** 2
**Presentation:** 3
**Significance:** 3
**Originality:** 4
**Overall Recommendation:** 4
**Confidence:** 5

**Summary:**

This paper proposes a deepfake detection method called Reliable Multi-View Evidential Learning.
The authors employ orthogonal projection to decompose entangled representations into two independent views: a Semantic View (high-level semantic consistency) and an Artifact View (low-level signals to expose manipulation artifacts).
Then they apply Evidential Deep Learning based on Dempster-Shafer (DS) theory for decision-level fusion, which models inter-view conflicts and provides calibrated uncertainty for reliable detection.
Extensive benchmarking demonstrates state-of-the-art performance on cross-domain datasets and high robustness to standard image degradations.

**Compliance With Llm Reviewing Policy:**

Affirmed.

**Final Justification:**

The paper studies an important concept of feature decoupling in deepfake detection, and the authors' response successfully clarified the theoretical and empirical basis for their "Divide-and-Conquer" strategy.  The authors have effectively addressed my concerns about source independence,  so I raised my score.

**Key Questions For Authors:**

1. Your framework relies heavily on Dempster-Shafer theory, which assumes evidence sources are independent. The paper claims that this independence is achieved by forcing geometric orthogonality ($\langle f_a, f_s \rangle = 0$). But I think in high-dimensional space, the fact that the inner product of two vectors is zero (or orthogonal) simply means that they are not linearly dependent. Given that $f_a$ is deterministically computed from $f_s$ through the VAE and projection steps, they are causally coupled. How do you justify the claim of source independence beyond mere geometric orthogonality?
2. The conflict-aware fusion excellently handles cases where the views disagree. However, what happens when an advanced model generates a structurally flawless deepfake that perfectly fits the VAE manifold? If both the semantic and artifact views falsely classify it as ``Real``, won't the Consensus Amplification cause the model to output a highly confident False Negative, completely neutralizing the uncertainty?
3. Geometric View Purification explicitly removes any component of the residual that is collinear with the semantic feature $f_s$. How do you ensure that this projection does not inadvertently erase high-level, semantic forgery artifacts?

**Limitations:**

yes

**Strengths And Weaknesses:**

Strengths：
1. The paper introduces a clever and explicit mathematical orthogonal projection approach to separate subtle fake artifacts from normal facial features.
2. The paper proposes Uncertainty-Aware Evidential Learning, which enables dynamic uncertainty quantification rather than forcing a blind, overconfident guess.
3. The paper presents reliable comparative experiments and detailed ablation experiments, and provides a thorough analysis of the experimental results.



Weaknesses：
1. The entire pipeline blindly trusts the initial CLIP encoder as a flawless semantic anchor. If an attacker applies invisible adversarial noise specifically designed to confuse CLIP, the initial feature extraction will be poisoned, causing the entire downstream separation and fusion process to collapse.
2.  The $\beta$-VAE is cascaded directly after the CLIP output semantic feature $f_s$, making the manifold-consistent feature $f_c$ (along with the subsequent residual $f_r$ and artifact feature $f_a$) strictly and solely dependent on $f_s$. In contrast, in traditional and rigorous Multi-view Learning, views are expected to be parallel and extracted directly from the source data.
3. The framework relies heavily on Dempster-Shafer theory, which strictly requires evidence sources to be statistically independent. However, because the artifact view is mathematically derived directly from the semantic view, they are inherently coupled, which undermines the statistical rigor of using this specific fusion method.
4. The paper did not conduct experiments on computational overhead and inference costs.


More minor details:

Page2-Right-L68:“Deepfake Detecion”-> Deepfake Detection

Page4-Left-L187: “real faces approximately reside on a compact manifold in the semantic feature space, while deepfakes deviate from it” Is there any reference for this assumption?

In Fig.2: It says “LoRA-tuned CLIP encoder” in the figure but “LoRA-adapted CLIP encoder” in the caption.

---

> ### Author Rebuttal · Authors · 2026-03-31
>
> **W1:**
>
> **1.  Robustness:** Our semantic stream attack experiments (`Tabs.I & II`) show that despite general degradation under targeted perturbations, our method achieves the highest performance and most stable degradation. Crucially, rising uncertainty provides an essential risk signal with robust predictions.
>
> **2. Practicality:** In real-world scenarios, our detection framework operates as a black box. Since attackers lack prior knowledge of the specific foundation model employed, the wide variety of available backbones makes precise semantic-stream-targeted attacks substantially harder in practice.
>
> **W2:**
> 1. Rather than directly instantiating classical parallel multi-view learning, ours is a constructed dual-view evidential framework. Instead of assuming independent raw views, we explicitly derive complementary, low-redundancy evidence streams from an initially coupled representation for effective evidential fusion.
> 2. `Table 2(a)` ablations of the main text support this: an ineffectively redundant branch would not yield consistent gains. Instead, this decomposition reliably enhances generalization, robustness & uncertainty estimation.
>
> **W4:**
>
> `Tab.III` shows that sharing the main visual backbone allows our lightweight modules to process compact features efficiently. This favorable trade-off yields stronger performance and reliable uncertainty estimates with only modest impact on deployment speed.
>
> **Q1&W3:**
>
> Consistent with the main text, we define independence not as strict statistical independence, but as a low-redundancy, highly complementary state that provides decorrelated, distinct evidence streams for effective and practical DS-style fusion. Below, we clarify how this state is achieved and utilized:
>
> **1. Effect of purification and decorrelation:** Purification shifts evidence streams from a strongly coupled regime to an approximately decorrelated state that is more suitable for evidence fusion. As shown in dependence diagnostics (`Tab.IV`), this sharp reduction in coupling persists across both intermediate features and final evidential heads.
>
> **2. Architectural synergy and fusion rationale:** Our use of DS-style fusion is motivated by distinct evidence streams alongside explicit conflict modeling. While orthogonal projection directly targets linear dependence, the reduction in nonlinear dependence (HSIC) reflects architectural synergy: β-VAE can suppress part of the nonlinear coupling via manifold reconstruction, while orthogonal projection removes the dominant shared semantic residual. Together, they substantially weaken both streams' primary information-sharing pathway.
>
> **Q2:**
>
> **1. Claim:** While our method does not theoretically eliminate this consensus error, the reviewer's stronger assumption does not automatically hold in our model: a structurally near-perfect deepfake, highly consistent with the VAE manifold, would cause the artifact branch to output strong Real evidence.
>
> **2. Mechanism:** In evidential learning, an absence of forgery traces produces high uncertainty, not strong Real support. Operating on purified representations lacking these traces, the artifact branch likely remains uncertain instead of consistently producing strong Real evidence. Consequently, DS fusion avoids strongly triggering consensus amplification. When the semantic branch supports Real alongside an uncertain artifact branch, the semantic-dominated result retains cross-view uncertainty, preventing the model from confidently amplifying a wrong conclusion.
>
> **3. Validation:** Uncertainty analysis (`Fig.3`) in the main text does not support overconfident false negatives as the primary failure mode. Misclassified samples concentrate heavily in high-uncertainty regions, and rejecting them consistently improves risk-coverage curves. Thus, while theoretically possible, empirical evidence does not support the practical concern of consensus amplification producing confident false negatives.
>
> **Q3:**
>
> **1. Claim:** Rather than globally eliminating semantic forgery artifacts, we prevent their redundant encoding in the artifact branch. Geometric View Purification solely targets the residual-derived artifact stream, while the intact $f_s$ branch directly drives final evidential predictions. Thus, discriminative semantic forgery cues remain available in the semantic stream.
>
> **2. Mechanism:** The projection removes only semantic-aligned residuals, likely duplicating information captured by $f_s$. Without this crucial separation, redundant evidence streams weaken conflict-aware fusion and increase shared errors.
>
> **3. Validation:** As `Table 2(a)` of the main text shows, *Ours* consistently outperforms baselines (A) & (C). This supports the interpretation that instead of suppressing useful forgery cues, the projection successfully reduces semantic leakage and improves cross-view complementarity.
>
> **Typos:**
>
> Thanks. We will fix the typos in the revision.
>
> > Tabs I-IV: https://anonymous.4open.science/r/add-F622.

---

> > ### Author Rebuttal · Reviewer_sFYT · 2026-04-02
> >
> > The authors have addressed my concerns.  I will raise my score.

---

> > > ### Author Response · Authors · 2026-04-02
> > >
> > > We are deeply grateful for your positive feedback and for raising the score. We wish you continued success in your research and all the very best in your life!

---

### Official Review · Reviewer_yXoQ · 2026-03-12

**Soundness:** 3
**Presentation:** 3
**Significance:** 2
**Originality:** 2
**Overall Recommendation:** 4
**Confidence:** 3

**Summary:**

This paper addresses cross-domain deepfake detection by decomposing entangled CLIP representations into a semantic view and an artifact view. The "Divide" phase uses a beta-VAE to reconstruct the semantic manifold, then projects the residual onto the orthogonal complement of the semantic direction to obtain purified artifact features. The "Conquer" phase maps each view to a Dirichlet distribution via evidential deep learning and fuses them using Dempster-Shafer combination. Experiments on cross-dataset (6 benchmarks) and cross-manipulation (DF40) protocols show consistent AUC improvements over 12 baselines.

**Compliance With Llm Reviewing Policy:**

Affirmed.

**Key Questions For Authors:**

Please refer to the weaknesses section. Also, Table 1 seems to compare methods with different backbone families. Could the authors clarify how much of the observed improvement over older baselines is attributable to the proposed method versus the backbone advantage?

**Limitations:**

yes

**Strengths And Weaknesses:**

### Strength

- The geometric orthogonal projection is clean and well-motivated. Defining artifacts as the orthogonal complement of the CLIP semantic direction is a simple inductive bias that directly addresses semantic leakage.

- The ablation study is thorough. Table 2 results isolate the artifact view, orthogonal projection, evidential loss, DS fusion, alignment loss, KL divergence, backbone scale, and pretraining strategy.

- Results are generally strong comparing to some SOTA baselines.

- The presentation is good and paper is generally easy to follow.

### Weaknesses

- The paper motivates DS-style evidential fusion by claiming that the semantic and artifact branches form independent evidence sources. However, although the method achieves approximate de-correlation via orthogonal projection, the features later pass through different learned heads (Eq. 5) that are jointly optimized. Joint training can reintroduce statistical dependence even if the input features are geometrically orthogonal. No diagnostic is provided to verify that the learned opinion distributions are actually independent.

- The originality claim could be positioned more modestly. The core geometric step is a standard sample-wise orthogonal projection that removes the residual component along the semantic feature direction, rather than a fundamentally new purification mechanism. Prior work has already used orthogonal subspace removal for disentanglement and debiasing, such as Effort’s SVD-based orthogonal decomposition. Generally speaking, the idea of orthogonal projection is well established and widely applied. The main novelty here seems to lie in the specific combination of per-sample residual projection with evidential multi-view fusion, not in the projection principle itself.

- No variance reporting across runs. All results appear to be single-seed and makes it a bit difficult to evaluate how significant the improvements are, especially for some tight margins shown in the result tables.

---

> ### Author Rebuttal · Authors · 2026-03-31
>
> **W1:**
>
> Thanks. Please refer to our detailed response to Reviewer sFYT, Q1&W3 on this point.
>
> * We additionally provide dependence diagnostics at both the feature level and the opinion-evidence level, using *Mean Absolute Full Correlation*, *Gaussian Mutual Information Proxy*, and *HSIC*. Specifically, *Mean Absolute Full Correlation* measures the overall linear correlation, the *Gaussian Mutual Information Proxy* approximates the amount of shared information between the two branches, and *HSIC* evaluates more general statistical dependence. For all three metrics, lower values indicate weaker dependence.
>
> * The results in `Tab.IV` show that all these metrics decrease sharply after purification, and importantly, the same trend persists at the final evidence level. ***This indicates that the reduction in dependence is not limited to the input features, but also carries through to the final opinions and evidence used for fusion***.
>
> > **Detailed experimental results** are available at our anonymous repository:  https://anonymous.4open.science/r/add-F622.
>
> ---
>
> **W2:**
>
> We thank the reviewer for this helpful suggestion. We will revise the manuscript to position the originality claim more modestly.
>
> ---
>
> **W3:**
>
> Thanks. We have added multi-run statistics in the revised manuscript and now report the results as mean ± std over 5 random seeds below. The results show that our method remains the strongest on the averaged cross-dataset and cross-manipulation evaluations, while also exhibiting low variance across runs. At the same time, we note that the original `Table 1` in the main text followed the common reporting protocol used in prior work, and most of the published baseline results there are also reported as single-run numbers. We therefore keep that format in `Table 1` in the main text to ensure direct comparability with existing methods.
>
> **Table I. Quantitative comparison on cross-dataset and cross-manipulation benchmarks (mean ± std over 5 seeds, best in bold).**
>
> | Methods | Venue | CDFv2 | DFD | DFDC | DFo | WDF | CDFv3 | Avg. | UniFace | BleFace | MobSwap | e4s | FaceDan | FSGAN | InSwap | SimSwap | Avg. |
> | :--- | :--- | :--- | :--- | :--- | :--- | :--- | :--- | :--- | :--- | :--- | :--- | :--- | :--- | :--- | :--- | :--- | :--- |
> | Effort | ICML'25 | 0.953 ± 0.003 | 0.963 ± 0.002 | 0.839 ± 0.006 | 0.975 ± 0.004 | 0.845 ± 0.004 | 0.848 ± 0.003 | 0.904 ± 0.004 | 0.961 ± 0.002 | 0.871 ± 0.002 | 0.951 ± 0.002 | 0.980 ± 0.003 | 0.922 ± 0.003 | 0.954 ± 0.003 | 0.932 ± 0.004 | 0.923 ± 0.003 | 0.937 ± 0.003 |
> | GenD | WACV'26 | 0.957 ± 0.003 | 0.964 ± 0.005 | 0.868 ± 0.003 | 0.987 ± 0.002 | 0.889 ± 0.002 | 0.851 ± 0.007 | 0.919 ± 0.004 | 0.971 ± 0.003 | 0.932 ± 0.003 | 0.974 ± 0.003 | 0.981 ± 0.002 | 0.940 ± 0.002 | 0.971 ± 0.003 | 0.941 ± 0.002 | 0.935 ± 0.003 | 0.956 ± 0.003 |
> | **Ours** | - | **0.976 ± 0.001** | **0.981 ± 0.002** | **0.881 ± 0.001** | **0.991 ± 0.002** | **0.910 ± 0.001** | **0.884 ± 0.002** | **0.937 ± 0.002** | **0.981 ± 0.002** | **0.961 ± 0.002** | **0.976 ± 0.002** | **0.990 ± 0.001** | **0.971 ± 0.002** | **0.987 ± 0.003** | **0.968 ± 0.003** | **0.963 ± 0.002** | **0.975 ± 0.002** |
>
> ---
>
> **Q1:**
>
> Thanks. We address this question according to the following points:
>
> **1.  Interpretation of Cross-Backbone Comparisons:** Comparisons across different backbone families cannot strictly isolate the contribution of the proposed method. Accordingly, comparisons against older methods with different backbones are mainly intended to show the overall empirical standing of the method, rather than a purely method-level gain. The attribution of gains to the proposed design should instead be judged primarily from the *Effort* / *GenD* / *Ours* same-backbone comparisons and the shared-backbone internal ablations.
>
> **2. Same-Backbone Evidence:** In particular, *Effort*, *GenD*, and *Ours* are all built on the same CLIP ViT-L/14 backbone, making this comparison substantially fairer for assessing the contribution of the method itself. Under this setting, `Table 1` of the main text shows that *Ours* still consistently outperforms both *Effort* and *GenD* on the cross-dataset and cross-manipulation protocols.
>
> **3. Internal Ablation Attribution:** As shown in `Table 2(a)` of the main text, (A) reflects the capability of the same CLIP ViT-L/14 backbone itself; on top of this baseline, introducing the second view in (B) already improves performance, indicating that the gain cannot be attributed to the backbone alone. Furthermore, if the unpurified residual variant (C) underperforms the purified artifact variant (*Ours*), this suggests that the improvement does not come merely from adding another branch or slightly increasing capacity, but from the proposed geometric purification design itself. ***Therefore, this same-backbone, same-protocol ablation chain provides a clearer attribution of the gains to the proposed method design rather than to backbone strength.***

---

> > ### Author Rebuttal · Reviewer_yXoQ · 2026-04-03
> >
> > I appreciate the author for addressing my questions. I will respectfully maintain my original score.

---

> > > ### Author Response · Authors · 2026-04-04
> > >
> > > We thank the reviewer for the valuable time and constructive comments. Your insightful feedback has significantly helped us improve the clarity and rigor of our manuscript.

---

### Official Review · Reviewer_oRxv · 2026-03-12

**Soundness:** 3
**Presentation:** 3
**Significance:** 3
**Originality:** 3
**Overall Recommendation:** 4
**Confidence:** 3

**Summary:**

The paper proposes a “Divide-and-Conquer” framework called Reliable Multi-View Evidential Learning. In the Divide phase, Geometric View Purification projects residuals onto the orthogonal complement of a LoRA-adapted CLIP semantic space to isolate artifact-sensitive representations. In the Conquer phase, Uncertainty-Aware Evidential Learning maps the purified views into Dirichlet-parameterized subjective opinions and fuses them via Dempster-Shafer theory, explicitly modeling epistemic conflict to produce calibrated uncertainty estimates.

**Compliance With Llm Reviewing Policy:**

Affirmed.

**Final Justification:**

As the clarifications provided in the rebuttal have resolved my initial concerns, I stand by my original positive recommendation.

**Key Questions For Authors:**

Please see weaknesses.

**Limitations:**

yes

**Strengths And Weaknesses:**

Strengths:

- Theoretical rigor and novelty in disentanglement. The authors provide a principled geometric foundation by explicitly defining artifacts as the orthogonal complement of the semantic subspace and enforcing ⟨f_a, f_s⟩ = 0 via vector projection (Equations 3–4). This is not a soft attention or rank-reduction heuristic (as in prior Effort or FDML works) but a hard geometric constraint that provably suppresses semantic leakage, directly addressing the Semantic Masking Effect. The derivation is mathematically clean and grounded in linear algebra, lending strong theoretical credibility.

- Principled uncertainty modeling. By replacing softmax with Subjective Logic + Dempster-Shafer combination (Equation 7), the method explicitly quantifies epistemic conflict C and redistributes belief mass only when views disagree. This is a substantial advance over deterministic fusion or naive averaging; the resulting fused uncertainty u serves as a reliable risk signal (validated by risk-coverage curves and ECE reduction from 0.16 to 0.07). The link between high-u samples and misclassifications is convincingly shown across FF++, CDFv2, DFDC, and DFD.

- Strong empirical superiority and comprehensive analysis. The method achieves consistent SOTA AUC (0.939–0.976 average) on six cross-dataset and ten cross-manipulation benchmarks, outperforming recent strong baselines (Effort, GenD, ProDet) by 1–3 points even under domain shift.

Weaknesses:
- The projection is performed at the global feature level. Sophisticated forgeries (e.g., localized blending, landmark-specific manipulations) may produce spatially heterogeneous artifacts; a single global f_s direction risks leaking local semantic cues into f_a. Region-adaptive or patch-wise projections are mentioned only as future work, yet this limitation could explain the remaining 1–2% gap on the hardest DFo and WDF subsets.

- All experiments use face-centric datasets (FF++, DF40, Celeb-DF variants). While the method claims “universal” forensic cues, no evaluation on general AI-generated images (e.g., Imagen, SDXL, or non-face scenes) or long-video temporal consistency is presented. Extension to temporal/cross-modal settings is acknowledged as future work but not quantified, weakening claims of broad applicability in the rapidly expanding AI-generated content landscape.

---

> ### Author Rebuttal · Authors · 2026-03-31
>
> **W1:**
>
> Thanks. We agree that complex forgeries are often spatially heterogeneous, and we highly value the reviewer's insight that region-adaptive or patch-wise purification is a meaningful extension.
>
> However, ***we do not believe that the current results support the interpretation that global projection is already the primary reason for the remaining gap on *DFo* and *WDF****. Our reasoning is twofold:
>
> **1.**  As shown in `Table 1` of the main text, despite utilizing a global Geometric View Purification, *Ours* still achieves the best performance on these two subsets. This demonstrates that the current global scheme is already highly robust and effective against localized forgery patterns, proving that the marginal performance gap cannot be attributed to this single design choice.
>
> **2.** In fact, we view the global projection as a ***first-order solution*** for disentangling features and reducing semantic leakage into the artifact branch. This design addresses the core challenge of the task directly and reasonably. While region-adaptive or patch-wise extensions might further push the performance upper bound, they represent more incremental optimizations. They are not necessary preconditions for our framework and do not diminish the essential nature of our primary contribution.
>
> ---
>
> **W2:**
>
> Thanks. ***The task setting of this paper is deepfake detection, which is typically defined around face-centric manipulated media***. Accordingly, standard benchmarks in this area are mainly built on datasets such as *FF++*, *DFDC*, *Celeb-DF*, and related variants, rather than general non-face AI-generated image detection. In this sense, our current experimental scope is consistent with the mainstream evaluation setting of deepfake detection. To further address this concern, we conducted two focused supplementary experiments as minimal validation:
>
> **1. Supplementary Experiment on General AI-Generated Portrait Detection**
>
> * Although the primary focus of our work is not general AI-generated content detection, we also conducted a supplementary experiment to probe the potential of the method beyond conventional face-swap datasets. Specifically, we evaluated it on *MidJourney* and *CollabDiff*, which belong to AI-generated portrait data rather than traditional video deepfake benchmarks.
>
> * The results show that direct transfer to these AIGC portrait domains remains challenging. However, after only 2 adaptation epochs, the target-domain AUROC substantially improves, from 0.689 to 0.915 on *MidJourney* and from 0.678 to 0.936 on *CollabDiff*, while causing only minimal change to *FF++* performance (0.997 and 0.990 after adaptation).
>
> * This suggests that the proposed forensic cues are not confined to the original training benchmark, but can also be adapted to the neighboring setting of AI-generated portraits.
>
> **2. Supplementary Temporal Aggregation Analysis**
>
> * Regarding temporal consistency, we agree that reporting video-level AUROC alone is not the same as explicitly quantifying long-video temporal extensibility. To partially address this concern, we conducted a focused supplementary temporal aggregation analysis under the same video-level AUROC metric on *FF++*, *CDFv2*, *DFD*, and *DFDC*.
>
> * The results show that standard mean aggregation achieves an average AUROC of 0.960, while inverse-uncertainty weighted mean improves this to 0.972, with the clearest gain on the most challenging *DFDC* setting (0.882 to 0.902). Here, standard mean aggregation assigns equal weight to all frame-level predictions, whereas inverse-uncertainty weighted mean gives larger weights to frames with lower evidential uncertainty, thereby emphasizing more reliable frame-level evidence during video-level aggregation.
>
> * We nevertheless keep standard mean aggregation in the main results for fair comparison with prior methods. We present this only as a minimal supplementary check showing that the model’s uncertainty output has practical temporal utility under video-level aggregation, rather than as a full validation of long-video temporal consistency or a new temporal module.

---

> > ### Author Rebuttal · Reviewer_oRxv · 2026-04-04
> >
> > Thanks for your reponse. I will keep my score.

---

> > > ### Author Response · Authors · 2026-04-04
> > >
> > > We deeply appreciate the reviewer’s thorough evaluation and constructive feedback. We will incorporate the necessary additions and revisions in the subsequent paper.

---

### Decision · Program_Chairs · 2026-04-30

**Decision:**

Accept (regular)

**Comment:**

## Summary of paper

This paper proposes a two-phase framework for cross-domain deepfake detection. The "Divide" phase decomposes CLIP features into semantic and artifact views via geometric orthogonal projection. The "Conquer" phase fuses these views using evidential deep learning with Dempster-Shafer theory, producing calibrated uncertainty estimates. The method achieves SOTA AUC across six cross-dataset and ten cross-manipulation benchmarks.

## Reviews and discussions

Three reviewers unanimously recommend weak accept (4/4/4).

The reviewers appreciate:
- clean geometric disentanglement
- principled uncertainty modelling
- strong empirical results
- thorough ablations

The main concerns raised were
- statistical independence assumption for DS fusion
- modest novelty of orthogonal projection itself
- lack of variance reporting

 All concerns were addressed during rebuttal - all three reviewers marked their concerns as fully resolved.

## Recommendation: accept